

# What flowers do we like? The influence of shape and color on the rating of flower beauty

Martin Hůla and Jaroslav Flegr

Department of Philosophy and History of Science, Faculty of Science, Charles University in Prague, Prague, Czech Republic

## ABSTRACT

There is no doubt that people find flowers beautiful. Surprisingly, we know very little about the actual properties which make flowers so appealing to humans. Although the evolutionary aesthetics provides some theories concerning generally preferred flower traits, empirical evidence is largely missing. In this study, we used an online survey in which residents of the Czech Republic ($n$ = 2006) rated the perceived beauty of 52 flower stimuli of diverse shapes and colors. Colored flowers were preferred over their uncolored versions. When controlling for flower shape, we found an unequal preference for different flower colors, blue being the most and yellow the least preferred. In the overall assessment of beauty, shape was more important than color. Prototypical flowers, i.e., radially symmetrical flowers with low complexity, were rated as the most beautiful. We also found a positive effect of sharp flower contours and blue color on the overall rating of flower beauty. The results may serve as a basis for further studies in some areas of the people-plant interaction research.

## INTRODUCTION

People across cultures find flowers beautiful. The aesthetic appreciation of flowers is manifested in many ways. We grow flowering plants in our apartments and gardens, horticulturists put much effort into breeding new types of ornamental flowers, and floral motifs are often present on paintings, fabrics, china or jewelry (*Appleton*, *1996*; *Eibl-Eibesfeldt*, *1989*). Flowers also serve as traditional and highly esteemed gifts (*Haviland-Jones et al.*, *2005*). This human attitude towards plants and flowers is known as phytophilia (*Eibl-Eibesfeldt*, *1989*).

Many aspects of people-plant relationships have been explored in past years, especially the effects of plants and flowers on the human psyche. Some researchers have suggested that the presence of plants positively affects mood (*Larsen et al.*, *1998*; *Shibata & Suzuki*, *2002*; *Haviland-Jones et al.*, *2005*) and attention (*Herzog et al.*, *1997*; *Kaplan & Kaplan*, *1995*; *Kaplan*, *1995*; *Lohr, Pearson-Mims & Goodwin*, *1996*; *Raanaas et al.*, *2011*; *Tennessen & Cimprich*, *1995*), reduces stress (*Cackowski & Nasar*, *2003*; *Grahn & Stigsdotter*, *2010*) and even decreases recovery time after surgery (*Ulrich*, *1984*).

Corresponding author
Martin Hůla,
martin.hula@natur.cuni.cz

The perceived beauty of flowers might influence the psychological benefits they provide to humans. It is thus reasonable to ask if there exist any common human flower preferences or whether the perceived beauty of flowers depends solely on individual taste. Although several studies targeted on best-selling flower products provide us with some data (*Behe et al.*, *1999*; *Yue & Behe*, *2010*), they have two major limitations. First, they typically focus on only one segment of products (such as geraniums or single stem cut flowers), so it is not possible to generalize their results. Moreover, these studies do not attempt to explain the causes of the observed preferences. Second, the studies combine the effect of morphological traits (color, number or size of the flowers on the plant etc.) with the effect of price, product packaging etc.

In our study, we address the issue from a more general perspective. We postulate that if there are any common preferences for different flower traits, they would have been shaped in the course of human evolution. We thus use theories and hypotheses from evolutionary aesthetics to predict which flower colors and shapes should generally be more preferred than others. Probably only one theory that explicitly mentions flowers has been published—the habitat selection theory of *Heerwagen & Orians* (*1993*), *Orians & Heerwagen* (*1995*) which we describe below. We also present other evolutionary hypotheses focused on general color and shape preferences and try to apply their outcomes to flowers. We then present the design and results of our study, which aimed to empirically test the validity of these hypotheses for flower preference. To increase the readability of the text, we discuss the preferred flower colors and shapes in two separate sections.

## Preferred flower colors

The habitat selection theory of Orians and Heerwagen regards flowers as important signs that could have helped our ancestors find a suitable habitat for living. The ability to choose a rich and safe habitat was essential for the survival of our ancestors, thus an innate preference for signs of such a habitat (and the avoidance of opposite signs) was highly adaptive. It is for this reason that we perceive these signs as beautiful. Flowers signal a rich environment and promise the presence of edible bulbs or fruits (*Heerwagen & Orians*, *1993*; *Orians & Heerwagen*, *1995*; *Pinker*, *1999*). Flower signs have to be visible from a distance, so we should mainly prefer their vivid and contrasting colors.

General color preference may also influence the beauty of many objects with the same color, including flowers. Green and blue colors could be preferred because they signal a rich and safe habitat (lush vegetation, water, clear sky). Brown or yellow are connected with barren land, drought, dead vegetation or feces and could be avoided (*Orians & Heerwagen*, *1995*, pp. 567–569; *Palmer & Schloss*, *2010*). On the other hand, edible fruits and nuts are often yellow or brown, so the predicted avoidance of these colors is somewhat dubious. Red color may signal edible fruits, sexual arousal or blood (*Humphrey*, *1980*). Red objects should be regarded as stimulating, but whether as beautiful is uncertain.

Some studies targeting the behavior of florist shop customers reported red and pink flowers as the most preferred and blue and yellow flowers as the least preferred (*Behe et al.*, *1999*; *Yue & Behe*, *2010*). A study examining the beauty of street flowers found equal preference for diverse flower colors (*Todorova, Asakawa & Aikoh*, *2004*). When people

rated their favorite color of a tree canopy, they most preferred red (*Kaufman & Lohr*, *2004*; *Heerwagen & Orians*, *1993*). However, in another study, a red canopy was the least preferred and blue had the highest rating (*Müderrisoğlu et al.*, *2009*).

People who rated the beauty of diverse birds appreciated the presence of blue and yellow coloration and overall lightness (*Lišková & Frynta*, *2013*). Similar results were found in the case of parrots (*Frynta et al.*, *2010*), while blue and green were the most preferred colors of pita birds (*Lišková, Landová & Frynta*, *2014*).

Studies examining overall color ranking have usually described blue and red as the top colors (blue was usually preferred slightly more by men and red by women) and yellow near the bottom (*Camgöz, Yener & Güvenç*, *2002*; *Ellis & Ficek*, *2001*; *Hurlbert & Ling*, *2007*; *Schloss, Strauss & Palmer*, *2013*; *Zemach, Chang & Teller*, *2007*). Color preferences also seem to be culturally dependent. For example, East Asian cultures have a preference for white color (*Saito*, *1996*), while members of the African Himba tribe highly esteem yellow and do not like blue (*Taylor, Clifford & Franklin*, *2013*).

*Palmer & Schloss* (*2010*) proposed the ecological valence theory, which integrates evolutionary and ontogenetic approaches in the research of human color preferences. The authors write that people should be attracted to colors they associate with salient objects they like and repulsed by colors associated with salient objects they dislike. They found a preference for blue color and a dislike for brown and dark shades of yellow. This pattern was consistent across several cultures (with slight variations). The authors thus concluded that some portion of color preference is probably universal while another portion is influenced by culture and individual experiences.

## Preferred flower shapes

The influence of flower shape on the perception of flower beauty was largely neglected by the theoretical and empirical works mentioned above. This is quite surprising, especially when we take into account the astonishing diversity of flower forms and the large number of studies documenting the importance of shape in the perception of beauty of many objects and organisms (see below).

Many authors have suggested that humans tend to aesthetically appreciate objects that are quickly recognizable and fluently processed by their brains. The presence of such objects assures easy orientation in the environment and rapid evaluation of its potential threats and benefits. Human attraction to these environments should be highly adaptive (*Humphrey*, *1980*; *Kaplan*, *1987*, *Kaplan 1988*; *Reber, Schwarz & Winkielman*, *2004*). Objects that are fluently processed tend to be symmetrical (*Enquist & Arak*, *1994*; *Enquist & Johnstone*, *1997*; *Jacobsen et al.*, *2006*; *Van der Helm & Leeuwenberg*, *1996*), prototypical (*Winkielman et al.*, *2006*), and moderately complex (*Reber, Schwarz, & Winkielman*, *2004*). Empirical research has confirmed that people prefer prototypical objects and animals (*Hekkert, Snelders & Wieringen*, *2003*; *Hekkert & Wieringen*, *1990*; *Reber, Schwarz & Winkielman*, *2004*).

Complexity influences the preference for objects (*Jacobsen et al.*, *2006*; *Reber, Schwarz & Winkielman*, *2004*), but not linearly. Studies have reported that objects with very low or very high complexity are preferred less than moderately complex ones (*Akalin et al.*, *2009*; *Hekkert & Wieringen*, *1990*). People dislike highly complex objects because they cannot be

easily and rapidly recognized and categorized, while objects with very low complexity are just boring. It is questionable whether we would observe an effect of boredom in the case of flowers, because even the simplest ones reach a certain base level of complexity.

Symmetrical objects are also considered beautiful (*Jacobsen & Höfel*, *2002*; *Jacobsen et al.*, *2006*; *Leder et al.*, *2004*). The processing fluency and the preference for objects increase with the number of their axes of symmetry (*Evans, Wenderoth & Cheng*, *2000*; *Tinio & Leder*, *2009*). This implies that radially symmetrical flowers should be preferred more than bilaterally symmetrical flowers. On the other hand, some researchers claim humans have a very strong preference for bilaterally symmetrical objects, which may be a by-product of the selection of partners (*Little & Jones*, *2003*) and the recognition of partners or enemies (*Johnstone*, *1994*; see also *Mithen*, *2003*). According to the habitat selection approach of *Heerwagen & Orians* (*1993*), the type of symmetry could provide information about the nutritive value of flowers. Bilaterally symmetrical flowers usually have more nectar than radially symmetrical ones and indicate richer habitats. For this reason, they should be regarded as more beautiful.

Recent studies have shown that people prefer round objects over objects with sharp contours (*Bar & Neta*, *2006*; *Leder, Tinio & Bar*, *2011*; *Silvia & Barona*, *2009*; *Westerman et al.*, *2012*). According to *Bar & Neta* (*2007*), this difference is due to the fact that objects with sharp contours evoke a subconscious feeling of danger and fear, which we inherited from our ancestors. However, another study suggested that the preference for round objects may be just a temporary fashion trend (*Carbon*, *2010*). Richard Coss argued that piercing forms (such as thorns, spikes, canines or horns) were certainly dangerous for our ancestors and even today arouse strong emotions, but not necessarily negative ones. Pointed forms may be strongly symbolic of power and mystery and could be aesthetically pleasing. One of his experiments showed that pedestrians and joggers actually approached plants with pointed leaves at a shorter distance than plants with round leaves. In another study, people rated silhouettes and patterns with sharp contours as more attractive than their rounded counterparts (*Coss*, *2003*).

## Relationship between shape and color

Research focusing on object recognition and representation has shown that shape plays the main role, but color is important too. When objects with typical colors (color diagnostic objects), such as a lime or carrot, are presented, a congruent color (orange carrot) facilitates performance while an incongruent color (blue carrot) causes performance to deteriorate (*Therriault, Yaxley, & Zwaan*, *2009*). A recent meta-analysis showed that color has some positive effect even on the recognition of objects without typical colors (non-color diagnostic objects). Color also had a stronger effect on natural objects than on artificial objects (*Bramão et al.*, *2011*). On the other hand, the relative weight of shape and color is context-dependent and can be influenced by both the nature of the object (for example fruit vs. animal) and also the task (categorization vs. motion evaluation) (*Scorolli & Borghi*, *2015*). If we assume that the beauty of an object is closely linked to the ease with which we can recognize and categorize it (see the section above), we should observe a

stronger effect of shape than color on the rating of flower beauty, although the presence of color should also serve to increase the perceived beauty of flowers.

### Aim of the study

The primary aim of this study was to determine which (if any) flower colors and shapes are more preferred than others. According to some of the mentioned theories from evolutionary aesthetics, flowers should be preferred because of their conspicuous colors. On the other hand, many studies have revealed that some shape properties influence the aesthetic appreciation of an object or a person. It is very likely that flower shape also plays a role in the assessment of the flower beauty. The literature is equivocal concerning the effect of some shape properties on preference (type of symmetry, sharp contours). Also, some of the well documented effects of shape on general object preference may be different when applied to flowers (complexity).

A second main objective of the study was to compare these theories with the empirical evidence and to evaluate the relative importance of color and shape. We wanted to answer the following questions: (1) Are there any general flower preferences? (2) Is the flower color more important than the flower shape? (3) Are some flower colors or shapes more preferred than others?

### Hypotheses

We proposed several hypotheses based on the research discussed above:
(1) We expected to find clear common flower preferences in our data set.
(2) We assumed that the presence of color would increase the rating of flower beauty.
(3) We expected to find differences in the beauty rating based on the specific flower color.
(4) We hypothesized that flower beauty would increase with perceived prototypicality,
(5) that moderately complex flowers would be considered more beautiful than those with very low or very high complexity, and
(6) that round flowers would be rated as more beautiful than those with sharp contours.
(7) Finally, we expected symmetry would play an important role in the evaluation of flower beauty, but it was not clear whether bilateral or radial symmetry should be more preferred.

## MATERIALS AND METHODS

To test our hypotheses, we conducted two independent online surveys targeted to the Czech population. Both surveys were based on the rating of photographs of flowers. First, we describe how we obtained the flower stimuli, then we present the design of both surveys. The dataset and flower stimuli are available at Figshare: https://figshare.com/s/7306f12659f68f7f3d9d.

### Flower Stimuli

We wanted to create a set of flower stimuli that would reflect the diversity of flower shapes and colors. However, it had to remain sufficiently small and easy to work with. For these reasons, we created a primary set of flowers that met the following conditions:

1. The plant is native to the Czech Republic.
2. The plant has no strong cultural connotations in the Czech environment (e.g., a rose is symbolic of love, etc.)
3. The size of the flower is between 1 and 4 cm in diameter.
4. Each flower can be clearly distinguished.

These conditions allowed us to reduce the immense number of flowering plants while maintaining a high morphological diversity. The flowers were not absolutely unknown or notoriously familiar to the respondents, as both of these situations could possibly lead to biased results. The flower size limit guaranteed that the shape of the real flowers could be normally seen with the naked eye. The preparation of the flower stimuli set also included the conversion of photographs to a single size, and it was desirable to keep the converted flower size close to the real one. The last condition eliminated possible problems with compact inflorescences, because it is arguable whether we should distinguish the appearance of single flowers in the inflorescence or treat the whole inflorescence as a single flower. The only exceptions to the last condition were the inflorescences of the aster family (*Asteraceae*). We included aster family members in the stimuli set because they are very common and the vast majority of people (laypersons) perceive their inflorescences as single flowers.

We found all the Czech flowering plant species in the Key to the Flora of the Czech Republic (*Kubát et al.*, *2002*). When the flowers met the inclusion criteria, we included them in the working flower set. In the case of genera with very similar species (e.g., *Rubus, Taraxacum*), we included the flower of just one species in the working set. The working set comprised flowers of 199 species, which we divided into 26 groups according to their shape. From each group we selected two flowers with different color (e.g., Fig. 1A) and added them to the final flower set (see Table 2).

We found freely available high quality photographs of each flower on the internet. To properly illustrate the true shape of the flowers, we used three photographs for each flower. These photographs were displayed together. The photograph in the center showed the flower from above (or *en face* in the case of bilaterally symmetrical flowers), while the photographs on the left and right sides depicted flowers that were turned slightly to the left and to the right, respectively (Figs. 1B and 1C).

We used Corel Photo Paint X7 to replace the original flower background by a neutral black color. The black background did not favor any flower (flowers are usually seen on a green, brown, grey or blue background) and provided enough contrast for the clear distinction of the flowers. We then centered the flowers and placed them in the same position, the top petal or tepal pointing directly upwards. Finally, we converted all of the flowers to the same size, optimal for displaying on most computer screens (flower = 150 pixels, flower + background = 200 pixels, the three photographs next to each other = 600 pixels). We also copied the final flower set and converted the photographs in it to a sepia tone (Fig. 1B). This new set was thus devoid of colors and helped us to test the influence of color on the rating of flower beauty. We did not use a conversion to a greyscale because grey photographs on a black background seemed somehow gloomy, which could negatively influence their rating.

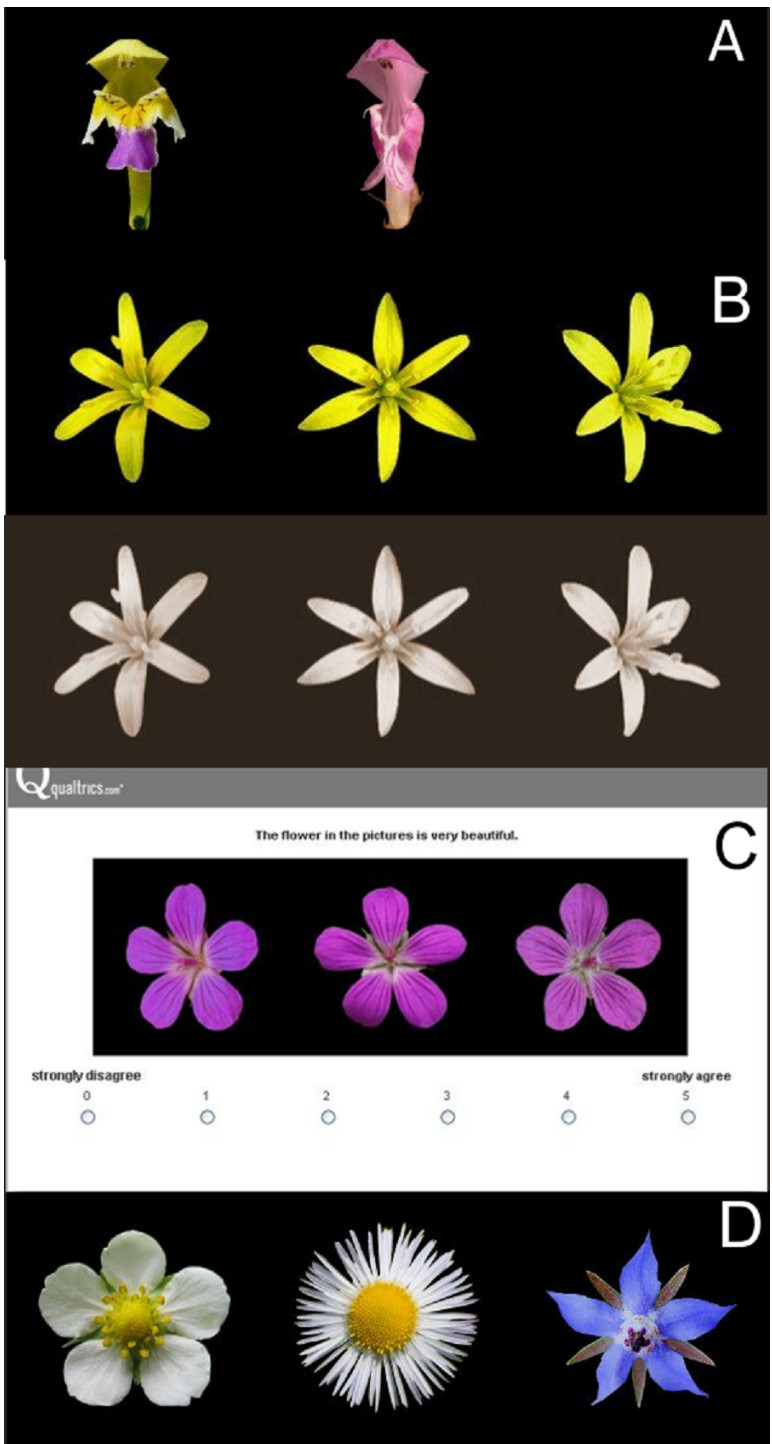

**Figure 1  Flower stimuli.** (A) examples of bilaterally symmetrical flowers with similar shape (left: *Galeopsis speciosa*, right: *Lamium maculatum*)—only the en face photographs; (B) colored flower stimulus and its sepia tone version (*Gagea lutea*); (C) example of a rating question setting (*Geranium palustre*); (D) Flowers with different angularity levels. Left: round (*Fragaria viridis*), center: mixed (*Erigeron annuus*), right: sharp (*Borago officinalis*).

The final set of flower stimuli consisted of 26 pairs of photographs, the flowers in each pair having a similar shape but a different color. There was also a sepia tone set of flower stimuli

## Determination of flower traits
### Symmetry
All flower stimuli in the set were symmetrical, but they differed in the type of symmetry. We distinguished radially symmetrical flowers (40 in total; e.g., Figs. 1B–1D) and bilaterally symmetrical flowers (12 in total; e.g., Fig. 1A), respecting the usual convention (for more details see, e.g., *Judd et al.*, *2002*, pp.: 66–67). We considered the inflorescences of the aster family (*Asteraceae*) as single radially symmetrical flowers.

### Angularity
We followed the approach of *Bar & Neta* (*2006*) when determining flower angularity. We divided flowers into three groups according to the curvature of their contours. There were flowers with round contours (21 in total), sharp contours (15 in total) and both round and sharp contours (16 in total). See Fig. 1D.

### Color
First we determined whether the flower had only a single color (22) or more colors (30). We also identified a dominant flower color (occupying at least 2/3 of the flower surface). To determine the dominant flower color, we cut a 30 x 30 pixels square (or its equivalent) from the area with the dominant color in each flower photograph. We then computed its average value in the hue-lightness-saturation (HLS) color space. The hue values correspond to the angles of a color wheel, where certain angles are associated with certain colors. We adopted the hue ranges published by *Newsam* (*2005*). To properly distinguish flower color, we had to avoid overlaps between the hue ranges of pink and purple. We set the range for purple to 270°–315° and the range for pink to 316°–350°. White, grey, and black colors can be defined by setting empirical thresholds of lightness (L) and saturation (S) values (*Lišková, Landová & Frynta*, *2014*; *Newsam*, *2005*). L and S can vary from 0 to 100. In our case, we defined white color as having L > 70 and S < 35. This combination of L and S values best matched the flowers perceived as white. With the described procedure, we defined the following color groups, which were later used in color preference analysis (the numbers in brackets represent the number of flowers within each group): white (14), yellow (8), blue (9), purple (8) and pink (7). Six flowers had a unique dominant color (*Hieracium aurantiacum*—orange, *Atropa bella-donna*—brown, *Arctium tomentosum*—green) or no dominant color (*Epipactis palustris*, *Galeopsis speciosa*, *Kickxia elatine*), and we excluded them from further color preference analysis.

## Survey design
Each survey consisted of a single questionnaire created in a Qualtrics environment.

In the first questionnaire the respondents rated a set of photographs of flowers by their beauty. The questionnaire also contained several sets of questions concerning basic information about the respondents, their attitude towards plants, color preferences and psychological characteristics.

Because the number of the flower stimuli was quite high (52 flowers in color and sepia tone), we decided to show each respondent only half of them (the first flower of each pair in color and in sepia tone, i.e., subset 1, or the second flower of each pair in color and sepia tone, i.e., subset 2). Although the flower stimuli in each subset remained the same, we randomized their display order. To prevent the respondents from rating the colored flower stimuli under the influence of the sepia tone stimuli and vice versa, we randomized the display order of the colored and sepia tone stimuli and also separated their rating by a set of questions.

For each flower stimulus, respondents expressed their agreement with the statement "The flower in the pictures is very beautiful." The respondents were choosing one point on a six point scale, where 1 meant "strongly disagree" and 6 meant "strongly agree" (Fig. 1C). The respondents moved to the next flower stimulus by clicking on the "next" button. Once the new flower stimulus appeared, it was no longer possible to change the rating of the previous ones (this fact was clearly explained before the start of the rating procedure).

In the second questionnaire the respondents rated the same set of photographs as in the previous questionnaire, but this time by their prototypicality and complexity. There was also a set of questions concerning basic information about the respondents and their attitude towards plants.

The second questionnaire contained fewer questions than the previous one, and it was also not necessary to rate the sepia tone flower stimuli. This allowed us to present each respondent with the whole set of flower stimuli (subset 1 and subset 2 together). We separated the rating of flower complexity and prototypicality by a set of questions and randomized the display order of each rating. The order of flower stimuli in each rating was also randomized. The rating instructions explained what flower complexity and prototypicality meant. For illustration, we also added two examples of the complexity and prototypicality rating of birds and butterflies. The rating procedure was the same as for the determination of flower beauty, but this time, the respondents expressed their agreement with the statements "This is how I imagine a complex flower." and "This is how I imagine a typical flower."

There was a break of several months between the start of the first and second surveys. We distributed the link to both surveys mainly via the Facebook group *Pokusní králíci* (Guinea Pigs; www.facebook.com/pokusnikralici, which is administered by the members of our laboratory (see *Flegr & Hodný*, *2016*; for details). The link was also displayed on other web pages; anyone could share the link.

Respondents gave their informed consent to the data collection by proceeding with the questionnaire (this fact was clearly explained on the first page of the questionnaire). Both surveys were completely anonymous. The research was approved by the IRB of the Charles University, Faculty of Science (Approval number: 2015/31).

## Characteristics of the respondents

The first questionnaire, in which flower beauty was determined, was completed by 2,006 people (1,484 women, 521 men and one person of unknown sex). Fifty percent of the respondents were between 23 and 33 years old; the youngest respondent was 12 and the

oldest 74. Forty-five percent of the respondents lived in towns with more than 50 thousand inhabitants. Fifty percent of the respondents had a college education, while twenty-eight percent of the respondents studied or worked in the field of biology.

The second questionnaire, in which flower complexity and prototypicality were determined, was completed by 582 people (427 women, 153 men and two people of unknown sex). Fifty percent of the respondents were between 25 and 38 years old. The youngest respondent was 10 and the oldest 88. Forty-three percent of the respondents lived in towns with more than 50 thousand inhabitants. Fifty-three percent of the respondents had a college education, while twenty-five percent of respondents studied or worked in the field of biology.

Color blind respondents were excluded from the data set.

The characteristics of the respondents were very similar in both questionnaires, and it is likely that many people completed both questionnaires. We can thus assume that the ratings from both questionnaires are mutually relevant and comparable.

## Statistical analyses

We analyzed the data using R software, version 3.1.3. The significance level $\alpha$ was set to 0.05 in all tests.

We computed the scores of the mean beauty, complexity and prototypicality rating of each flower from all respondents. The scores could theoretically vary from 1 to 6 points. The score of flower beauty represented the dependent variable. In the color preference analysis, we computed the difference between the beauty scores of each colored flower and its sepia tone version. The difference could theoretically vary from $-5$ to $+5$ points. This difference then served as the dependent variable.

To determine the relationship between beauty, complexity and prototypicality, we used Pearson's correlation test (for normal distributions) or Spearman's rank correlation. We used the partial Kendall's correlation (R package 'ppcor') when it was necessary to filter the effect of a confounding variable. When comparing the means of two groups, we used Student's $t$-test (for normal distributions) or Wilcoxon's rank sum test. We also created general linear models to determine the relative importance of flower traits in the rating of flower beauty. We simplified the initial full model by stepwise backward elimination in order to ensure that the final reduced model could not differ significantly from the initial full model.

## Comparison of stimuli subsets

Each stimuli subset was rated by one-half of the respondents. We divided the stimuli into 26 pairs with similar (not identical) shapes and different colors. We allocated one member of each pair to subset 1 and one member to subset 2. We wanted to ensure that the flower stimuli in each pair had similar beauty scores when we controlled for the effect of color. We used a paired $t$-test to compare the beauty scores between the members of each pair (sepia tone version); no significant differences were found (mean difference $= 0.017$ point, 95% CI [$-0.18$–$0.21$], $t = 0.18$, $df = 25$, $p = 0.86$, Cohen's d $= 0.035$). We found a strong positive correlation between the beauty scores of subset 1 and subset 2 ($r = 0.63$,

**Table 1  The influence of display order on the beauty scores.**

| | color | | sepia | |
|---|---|---|---|---|
| | mean | var | mean | var |
| color first | 4.08 | 0.22 | 3.89 | 0.36 |
| sepia first | 4.14 | 0.27 | 3.99 | 0.3 |

Notes.

Color, colored stimuli set; sepia, sepia tone stimuli set; color first, the colored stimuli set was displayed first; sepia first, the sepia tone stimuli set was displayed first; mean, mean beauty score; var, variance of the beauty score.

95% CI [0.32–0.82], $t = 4.00$, $df = 24$, $p < 0.001$). For this reason, we pooled the data from both subsets and analyzed them together.

Exposure to the colored images could have influenced the ratings of the sepia tone images or vice versa. Therefore, one part of the participants first rated the sepia and then the colored images, while the second part of the participants first rated the colored and then the sepia images. We calculated the mean beauty scores and variances of the flower stimuli for each display option  (Table 1).

The mean beauty scores of the sepia tone flowers were lower when they were displayed after the colored flowers than when they were displayed before the colored flowers ($t = -4.50$, $df = 51$, $p < 0.001$, mean difference $= -0.096$, 95% CI [–0.14––0.05]). The variance followed the opposite trend.

Similarly, the mean beauty scores of the colored flowers were lower when they were displayed after the sepia tone flowers than when they were displayed before the sepia tone flowers ($t = -2.98$, $df = 51$, $p = 0.0044$, mean difference $= -0.052$, 95% CI [–0.087––0.017]). Again, the variance followed the opposite trend.

We took these findings into account in the subsequent analyses.

## RESULTS

### Flower color

We used a paired $t$-test to compare the mean beauty rating of colored and sepia tone flowers. Colored flowers had a significantly higher rating than the sepia tone ones (mean color $=$ 4.13, sd $= 0.50$; mean sepia $= 3.98$, sd $= 0.56$; mean difference $= 0.15$, 95% CI [0.07–0.22], $t = 4.02$, $df = 51$, $p < 0.001$, Cohen's d $= 0.56$). There was a strong positive correlation between the beauty rating of colored flowers and their sepia tone versions ($\rho = 0.85$, 95% CI [0.75–0.91], $S = 3609.1$, $p < 0.001$).

To determine whether the dominant flower color (hue) influenced its beauty rating, we created a general linear model in which the difference between the beauty score of each colored flower and its sepia tone version was the dependent variable. As explanatory variables we used the flower traits that could theoretically influence this difference. These were: dominant flower color (hue), lightness of the dominant flower color, saturation of the dominant flower color, number of colors in each flower, and flower prototypicality, symmetry and angularity. The initial full model (adjusted $R^2 = 0.56$) showed a significant effect of dominant flower color and symmetry. However, the final model (see Table 3) consisted of only one explanatory variable—the dominant flower color (hue)—and was

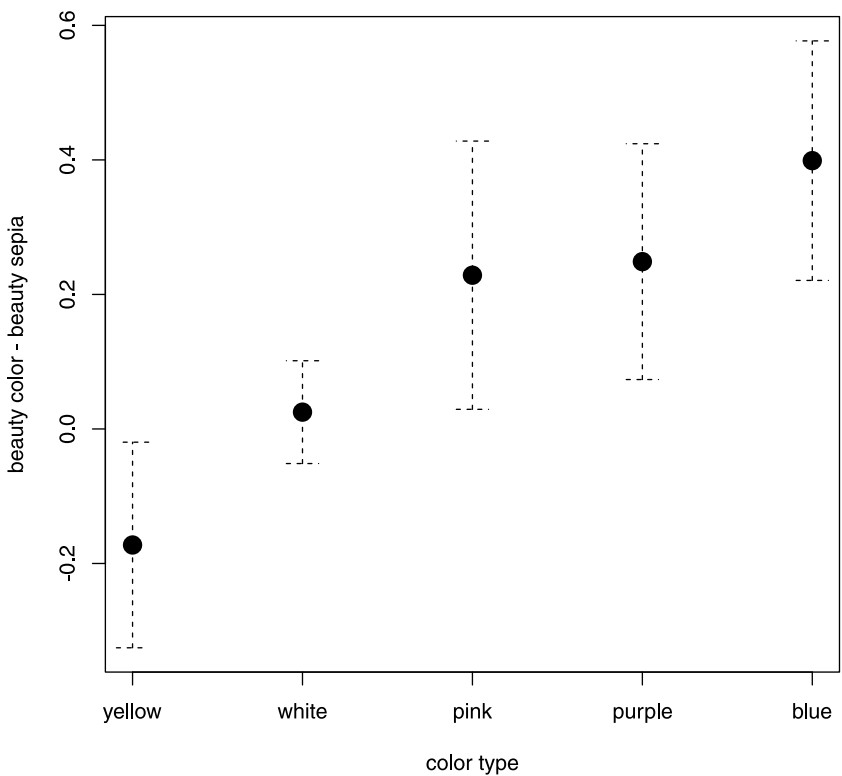

**Figure 2** **Effect of flower color on the estimation of beauty.** X axis: different flower colors (hues), Y axis: difference between the mean beauty rating of the colored flowers and their sepia tone versions. Error bars represent the 95% CI.

highly significant (adjusted $R^2 = 0.49$, $F_{4,41} = 11.91$, $p < 0.001$). Tukey–Krammer's post hoc test revealed that blue color was the most preferred. The mean difference between the rating of blue flowers and their sepia tone versions was 0.40. Blue was followed by purple (0.25 point) and pink (0.23 point). White color had no significant effect, and yellow flowers were rated even worse than their sepia tone versions (−0.17 point). See Fig. 2 and Table 4 for details.

To test the influence of the display order of the stimuli (colored set first vs. sepia tone set first), we applied the same model to the group in which the sepia tone stimuli were shown first and to the group in which the colored stimuli were shown first. In the "sepia-first" group, the final model only slightly differed in the values of the estimates (see Table 5 and Table 6). In the "color-first" group, however, the final model also revealed a significant positive effect of bilateral symmetry (apart from the effect of the dominant color). See Table 7 and Table 8.

## Beauty scores and flower traits

We determined the relationship between the scores of flower beauty, complexity and prototypicality. There was a significant positive correlation between the beauty and prototypicality scores ($\rho = 0.75$, $S = 36660.39$, $p < 0.001$; Fig. 3B). We found a significant negative correlation between the flower beauty and complexity scores ($\rho = -0.56$,

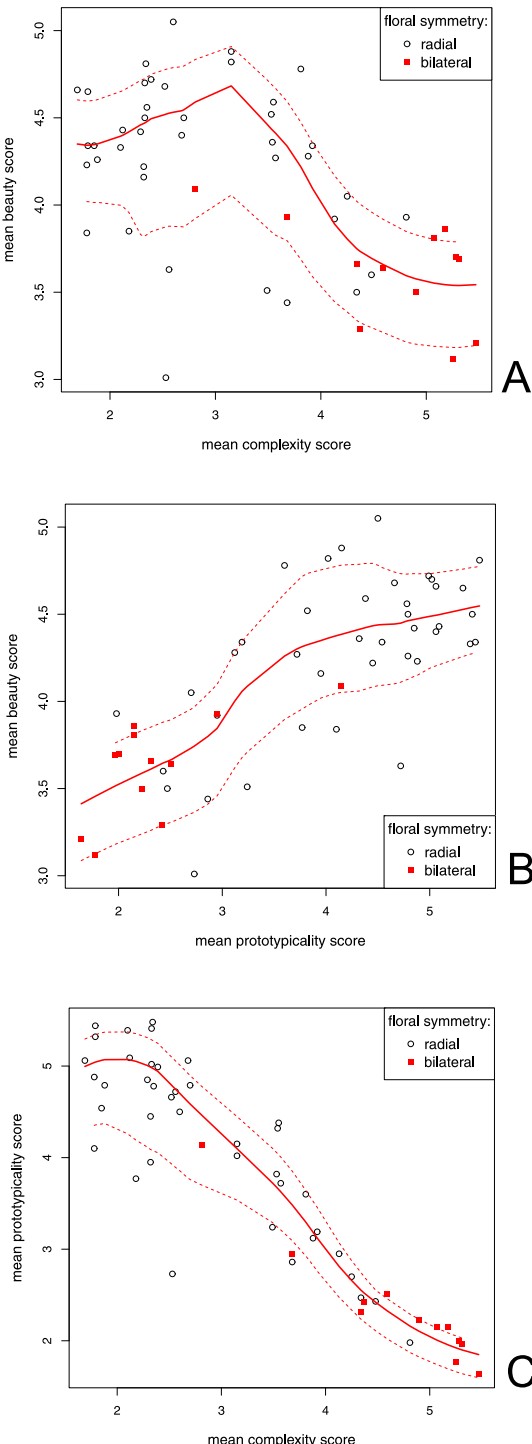

**Figure 3** **Correlation between the mean beauty, complexity and prototypicality ratings.** Each variable could vary from 1 (least beautiful/complex/prototypical) to 6 (most beautiful/complex/prototypical). A LOESS fitted line is shown (full line). Dashed lines represent the function spread ($\pm$SD) (A) Correlation between the beauty and complexity scores. $S = 36660.39$, $p < 0.001$, $\rho = -0.56$, 95% Cl $[-0.72$–$0.34]$; (B) Correlation between the beauty and prototypicality scores. $S = 5750.47$, $p < 0.001$, $\rho = 0.75$, 95% $[0.60$–$0.85]$; (C) Correlation between the prototypicality and complexity scores. $t = -15.61$, $df = 50$, $p < 0.001$, $r = -0.91$, 95% CI $[-0.95$–$-0.85]$.

$S = 5750.47$, $p < 0.001$; Fig. 3A). There was, however, a very strong negative correlation between the complexity and prototypicality scores ($r = -0.91$, $t = -15.61$, $df = 50$, $p < 0.001$, 95% CI $[-0.95–0.85]$; Fig. 3C). For this reason, we also computed the Kendall's partial correlation between the beauty and complexity scores, when controlling for prototypicality (and vice versa). There was still a significant positive correlation between the beauty and prototypicality scores when we excluded the effect of complexity ($z = 4.13$, $df = 50$, $p < 0.001$, $\tau = 0.40$), but there was no correlation between the beauty and complexity scores when we excluded the effect of prototypicality ($z = 0.41$, $df = 50$, $p = 0.68$, $\tau = 0.040$).

We used a Wilcoxon rank sum to determine the differences in the complexity and prototypicality scores of bilaterally and radially symmetrical flowers. To reveal the difference in beauty scores between bilaterally and radially symmetrical flowers, we used a two sample $t$-test. Radially symmetrical flowers scored higher in beauty (mean difference = 0.65 points, 95% CI $[0.37–0.93]$, $t = 4.65$, $p < 0.001$, Cohen's d = 2.00) and prototypicality (median bilateral = 2.19 points, median radial = 4.42 points, $W = 447.5$, $p < 0.001$, Hodges-Lehmann estimator = 2.02, 95% CI $[1.25–2.56]$). Bilaterally symmetrical flowers had higher scores in complexity (median bilateral = 4.99 points, median radial = 2.55 points, $W = 30$, $p < 0.001$, Hodges-Lehmann estimator $= -1.93$, 95% CI $[-2.61–1.26]$). All significant results remained significant also after performing the Bonferroni correction for multiple tests.

To determine the relative importance of different flower traits for rating their beauty, we created a general linear model in which the flower beauty scores served as the dependent variable. We wanted to include the dominant flower color (hue) in the model. At the same time, we also wanted to use the information contained in those flowers with a unique or uncertain dominant color (hue), which were deleted from the dataset in the previous color analysis. For this reason, we converted the factor variable dominant color (hue), which had five levels, into five binary variables (with levels of *no* and *yes*): white, yellow, purple, pink and blue. We also used the same procedure with the variable angularity. This step allowed us to gain information from the whole dataset and avoid reducing the degrees of freedom. As further explanatory variables we used the following flower traits: prototypicality, the number of colors in each flower, symmetry, lightness of the dominant flower color and saturation of the dominant flower color (or the most common color in the case of flowers with an uncertain dominant color). We did not include complexity in the model because of its very strong correlation ($r = -0.91$) with flower prototypicality.

The initial full model ($R^2 = 0.75$, adjusted $R^2 = 0.68$) revealed a significant effect of prototypicality, blue color, angularity and saturation. The final reduced model (Table 9) confirmed only the effect of prototypicality, blue color and sharp contours (adjusted $R^2 = 0.70$, $F_{3,48} = 39.81$, $p < 0.001$). All three of these variables had a significant positive effect on the mean flower beauty rating. The most important was prototypicality, followed by blue dominant color and sharp flower contours (Table 10).

As a control, we also created another linear model in which the flower hues were represented as levels of a single factor variable and the flowers with a unique or uncertain dominant color were deleted from the dataset. The final reduced model was very similar

to the model in which no flowers were excluded from the data set (adjusted $R^2 = 0.64$, $F_{7,38} = 12.50$, $p < 0.001$), and it contained the same variables with similar significant effects (prototypicality: estimate $= 0.32$, 95% CI [0.23–0.42], $t = 7.02$, $p < 0.001$; dominant blue color: estimate $= 0.35$, 95% CI [0.09–0.62], $t = 2.72$, $p = 0.010$; sharp contours: estimate $= 0.30$, 95% CI [0.076–0.53], $t = 2.70$, $p = 0.010$).

To test the influence of the display order of the stimuli (colored set first vs. sepia tone set first), we applied the same models to the group where the sepia tone stimuli were shown first and to the group where the colored stimuli were shown first. In both groups, the models only slightly differed in the estimate values (see Tables 11–14).

## DISCUSSION

We found that the presence of color generally slightly increased the beauty rating of flowers. When we compared colored and sepia tone versions of the same flowers, we found significant differences in the effects of specific colors. Blue was the most preferred, followed by pink and purple. As expected, white flowers did not differ from their sepia tone versions in their ratings, because both versions looked very similar. Yellow flowers were rated as less beautiful than their sepia tone versions. We were not able to measure the effect of red because only one genus (*Papaver*) native to the Czech Republic typically has red flowers.

Our results partly correspond with the habitat selection theory (*Heerwagen & Orians, 1993*) and also with the ecological valence theory (*Palmer & Schloss, 2010*). Both theories suggest people like blue color, which is typically related to clear sky or water, and tend to dislike brown and some shades of yellow because they are related to feces, death, vegetation or drought. The habitat selection theory links color preferences to the signs of the environment that were crucial for the survival of our ancestors. It assumes that our color preferences are a heritage of the past, hardwired in our brains. The ecological valence theory also recognizes inborn preferences but argues that these preferences can change during the course of an individual's life. It states that our color preferences are influenced by the valence of typically colored objects in our surroundings.

According to empirical research on the perceived beauty of simple colors (*Camgöz, Yener, & Güvenç, 2002*; *Ellis & Ficek, 2001*; *Hurlbert & Ling, 2007*; *Schloss, Strauss & Palmer, 2013*; *Zemach, Chang & Teller, 2007*); and tree canopies (*Müderrisoğlu et al., 2009*), blue is the most and yellow the least attractive color. A preference for blue was also reported for pita birds, which are very similar in shape but differ in coloration (*Lišková, Landová & Frynta, 2014*). We can assume that the general human color preference (as determined in American and European populations) also applies to flowers.

We must point out, however, that the yellow color (least preferred) in our set of stimuli was saturated. It is evident that clear yellow is more related to the sun or ripe fruits than to dead vegetation or drought. In our opinion, the habitat selection theory cannot fully explain the dislike of saturated yellow. Studies based on the ecological valence theory reported a low preference only for dark shades of yellow, whereas saturated yellow had an average preference. If we follow the assumptions of the ecological valence theory, we could argue that the Czech population tends to particularly dislike salient objects that typically have a

saturated yellow color. This in turn could lead to a low general preference for saturated yellow and explain the observed dislike of yellow flowers.

There is no agreement on the effect of lightness on the beauty rating of objects and organisms. *Lišková & Frynta*, (*2013*) stated that the beauty rating of birds increased with the overall lightness of their coloration. Schloss and colleagues (*2013*) found that lightness had no effect on the rating of color squares, a negative effect on the rating of small objects (e.g., t-shirt, pillow) and a positive effect on the rating of large objects (walls). We found no effect of lightness on the beauty rating of flowers. These differences in results may be caused by the use of different procedures to determine the degree of lightness and also by differences in stimuli presentation. It is also probable that the relative importance of lightness is context dependent.

It is important to note that although there were differences in flower color preference, they had only a minor effect when compared to the importance of flower shape. Only the presence of blue color significantly affected the beauty rating of flowers with diverse shapes. This relative unimportance of color was also found in the beauty rating of birds, whereas their shape (such as the length of the tail) had the major effect. However, blue and yellow colors also affected the perceived beauty of birds (*Frynta et al.*, *2010*; *Lišková & Frynta*, *2013*). Our results argue against the habitat selection theory, which suggests people like flowers mainly because of their vivid colors. According to our findings, flower market surveys might consider paying more attention to the shape of their products when trying to explore the preferences of their customers.

We report a very close relationship between the perceived flower prototypicality, complexity and type of symmetry. We expected to find a negative correlation between the prototypicality and complexity scores, but not as strong as our results actually indicate ($r = -0.91$). It would be helpful to compare the perceived complexity scores with some objective measurements. Unfortunately, it is very difficult to find an objective measurement method that could be applied to flowers with such a diversity of shapes.

The observed relationship between the flower beauty and complexity scores was close to an inverse U shape (Fig. 3A). This finding is in accord with previous research (*Akalin et al.*, *2009*; *Hekkert & Wieringen*, *1990*). Overly simple objects are usually described as boring, while very complex objects are difficult to process, which could explain their low preference (*Reber, Schwarz & Winkielman*, *2004*). We can see, however, that people still rated very simple flowers as quite beautiful, especially when compared with their rating of very complex flowers. This finding supports our assumption that flowers always have some base level of complexity, which assures they are never too boring to appreciate.

Bilaterally symmetrical flowers scored very low in prototypicality and very high in complexity. It is true that bilaterally symmetrical flowers are less common in the Czech Republic (and also worldwide). They often have fused floral parts and are highly three dimensional, so it might be difficult to describe their shape. These facts may account for their low prototypicality and high complexity scores.

We observed large differences in beauty scores between bilaterally and radially symmetrical flowers (radially symmetrical flowers scored higher). This supports the hypothesis that more axes of symmetry should lead to more fluent processing of the object

and its higher preference (*Evans, Wenderoth & Cheng*, *2000*). The results of our study go against the assumptions of *Heerwagen & Orians* (*1993*), who expected to find higher preference for bilaterally symmetrical flowers because they signaled richer habitats than radially symmetrical flowers. Our findings may quite paradoxically support the hypothesis that people tend to associate bilateral symmetry with human faces and bodies or with animals (*Little & Jones*, *2003*; *Mithen*, *2003*), but they are in opposition to its predicted outcome—a preference for bilateral symmetry. Bilaterally symmetrical flowers might be difficult to categorize. Their confounding animal- or even humanlike appearance might lead to their low preference. Anecdotal evidence supports this hypothesis. When we asked some of the raters about the flower stimuli, they often described the bilaterally symmetrical flowers as menacing and bizarre. The flowers reminded them of open mouths, snake heads and even aliens.

Partial correlations and the linear models also revealed that prototypicality encompasses both complexity and symmetry and is the main predictor of flower beauty. When we included prototypicality in our model, complexity and symmetry had no effect on flower beauty. Prototypical flowers had high beauty and low complexity ratings and were radially symmetrical.

Angularity also had a significant effect on the beauty scores. It turned out that sharp contours positively affected the flower beauty scores, while mixed contours had no effect. Our results disagree with those of some recent studies (*Bar & Neta*, *2006*; *Silvia & Barona*, *2009*), perhaps due to the different rating methods used. Previous research used forced choice methods in which the participants had to choose between two similar objects with different contours (e.g., sofa, watch, flower, rectangle etc.). In our study, each flower was rated separately, and we created no matching pairs with different levels of angularity. We have already mentioned that in some cases, sharp contours could be aesthetically pleasing (*Coss*, *2003*), thus we cannot dismiss the possibility that a preference for roundness and an avoidance of sharpness are context-specific and do not apply to flowers.

The display order of the stimuli (colored set shown first vs. sepia tone set shown first) affected the results of the linear model that examined the influence of color on flower preference. In the "sepia-first" group, only the effect of flower color was revealed. In the "color-first" group, we observed the effect of flower color and a positive effect of bilateral symmetry. In other words, the difference between the beauty scores of the colored and sepia tone versions of the same flower was greater for bilaterally symmetrical flowers than for radially symmetrical flowers.

In contrast to the radially symmetrical flowers, the bilaterally symmetrical flowers were generally rated as very complex and atypical. We can thus assume that they were difficult to recognize and categorize. Inês Bramão and her colleagues (*2011*) found that the recognition of non-color diagnostic objects (flowers are such objects) was facilitated when color was present. According to a number of works mentioned previously, an increase in processing fluency (the ease with which our brain can recognize objects) also increases the preference of the perceived object. This may explain the observed relative importance of color for rating the beauty of bilaterally symmetrical flowers when compared to radially symmetrical

ones. It is unclear, however, why we observed this effect only in the "color-first" group and not in the "sepia-first" group.

## Limitations and prospects

We have already mentioned some limitations of our study. First, we cannot overly generalize the results because the survey was conducted only on a non-representative (although quite large) sample of the Czech population. Cultural and individual differences in the evaluation of flower beauty (such as the effect of age, education or level of expertise) should certainly be explored in the future.

The display order of the stimuli (colored set first vs. sepia tone set first) influenced the beauty rating. It did not markedly affect the outcomes of most of the analyses, but the potential importance of the display order should be kept in mind when designing future studies.

Another limitation of our study was the fact that the respondents rated only photographs of single flowers. We should design an experiment in which real flowers would be rated and compare the results to those of the present study. A growing body of research shows that the human recognition and categorization of objects and entities is closely linked to, and often facilitated by, interaction with the environment through a sensory-motor activity (*Morlino et al.*, *2015*; *Scorolli & Borghi*, *2015*; *Smith*, *2005a*; *Smith*, *2015b*). It would certainly be beneficial to take this into account in the research of flower beauty. We could, for example, ask people to touch the flowers or to imagine that they pick/smell/give/receive the displayed flowers and then have them rate their beauty.

The relationship between prototypicality, complexity and symmetry is worthy of greater interest, not only in the case of flowers, but also in general. Attention should also be paid to the effect of red color on the rating of flower beauty, possibly by repeating the study with a more heterogeneous set of flowers not native to the Czech Republic.

The existence of unequal preferences for diverse flower traits opens an interesting question concerning the effects of flowers and plants on human health and performance. We should explore whether the effects of flowers and plants on human well-being change with their perceived beauty.

## CONCLUSION

Our research provides some empirical evidence for the evolutionary theories concerning the aesthetic evaluation of flowers. The results suggest that people share common preferences for certain flower traits. It seems that perceived flower beauty is influenced by flower color. Blue color increased and yellow decreased the perception of flower beauty, which is partially in accordance with the habitat selection theory of Heerwagen and Orians and also with the ecological valence theory. However, our results also showed that flower shape is the dominant feature in the beauty rating, substantially more important than color, and that prototypicality has a major positive effect on the perceived beauty of flowers.

## APPENDIX 1. LIST OF FLOWER STIMULI

List of Flower Stimuli is available in Table 2.

Hůla and Flegr (2016), *PeerJ*, DOI 10.7717/peerj.2106

**Table 2** List of flower stimuli.

| Scientific name | English name | Family | Pair | Symmetry | Beauty-color | Beauty-sepia | Complexity | Prototypicality | Angularity | Color |
|---|---|---|---|---|---|---|---|---|---|---|
| *Alisma plantago-aquatica* | common water-plantain | *Alismataceae* | 1 | radial | 3.51 | 3.64 | 3.49 | 3.24 | mixed | pink |
| *Sagittaria sagittifolia* | arrowhead | *Alismataceae* | 1 | radial | 4.16 | 3.88 | 2.32 | 3.95 | round | white |
| *Anthericum liliago* | st Bernard's lily | *Asparagaceae* | 2 | radial | 4.34 | 4.32 | 1.85 | 4.54 | sharp | white |
| *Gagea lutea* | yellow star of Bethelem | *Liliaceae* | 2 | radial | 4.26 | 4.5 | 1.88 | 4.79 | round | yellow |
| *Anoda cristata* | spurred anoda | *Malvaceae* | 3 | radial | 4.23 | 4.33 | 1.78 | 4.88 | round | purple |
| *Linum austriacum* | asian flax | *Linaceae* | 3 | radial | 4.66 | 4.29 | 1.69 | 5.06 | round | blue |
| *Dianthus superbus* | fringed pink | *Caryophyllaceae* | 4 | radial | 3.93 | 4.06 | 4.81 | 1.98 | sharp | white |
| *Lychnis flos-cuculi* | ragged-robin | *Caryophyllaceae* | 4 | radial | 3.5 | 3.21 | 4.34 | 2.47 | sharp | purple |
| *Dianthus carthusianorum* | carthusian pink | *Caryophyllaceae* | 5 | radial | 4.68 | 4.45 | 2.52 | 4.66 | sharp | pink |
| *Mycelis muralis* | wall lettuce | *Asteraceae* | 5 | radial | 4.22 | 4.3 | 2.32 | 4.45 | sharp | yellow |
| *Aster alpinus* | alpine aster | *Asteraceae* | 6 | radial | 4.81 | 4.66 | 2.34 | 5.48 | round | blue |
| *Erigeron annuus* | annual flea-bane | *Asteraceae* | 6 | radial | 4.5 | 4.32 | 2.33 | 5.41 | mixed | white |
| *Eruca sativa* | salad rocket | *Brassicaceae* | 7 | radial | 3.01 | 3.05 | 2.53 | 2.73 | round | white |
| *Lunaria annua* | annual hon-esty | *Brassicaceae* | 7 | radial | 3.84 | 3.2 | 1.78 | 4.1 | round | purple |
| *Erythronium dens-canis* | dogtooth vio-let | *Liliaceae* | 8 | radial | 4.05 | 3.76 | 4.25 | 2.7 | sharp | purple |
| *Lilium martagon alba* | white Turk's cap lily | *Liliaceaea* | 8 | radial | 4.28 | 4.31 | 3.88 | 3.12 | mixed | white |
| *Euphrasia rostkoviana* | eyebright | *Orobanchaceae* | 9 | bilateral | 3.81 | 3.78 | 5.07 | 2.15 | mixed | white |
| *Melittis melissophyllum* | bastard balm | *Lamiaceae* | 9 | bilateral | 3.29 | 3.12 | 4.37 | 2.42 | round | pink |
| *Anemone ranunculoides* | yellow anemone | *Ranunculaceae* | 10 | radial | 4.34 | 4.52 | 1.79 | 5.44 | round | yellow |
| *Fragaria viridis* | wild straw-berry | *Rosaceae* | 10 | radial | 4.33 | 4.34 | 2.1 | 5.39 | round | white |
| *Galeopsis speciosa* | large-flowered hemp nettle | *Lamiaceae* | 11 | bilateral | 3.69 | 3.24 | 5.31 | 1.97 | mixed | NA |

Hůla and Flegr (2016), *PeerJ*, DOI 10.7717/peerj.2106

**Table 2** (*continued*)

| Scientific name | English name | Family | Pair | Symmetry | Beauty-color | Beauty-sepia | Complexity | Prototypicality | Angularity | Color |
|---|---|---|---|---|---|---|---|---|---|---|
| *Lamium maculatum* | spotted dead-nettle | *Lamiaceae* | 11 | bilateral | 3.12 | 2.68 | 5.25 | 1.77 | round | pink |
| *Convolvulus arvensis* | field bindweed | *Convolvulaceae* | 12 | radial | 3.85 | 3.91 | 2.18 | 3.77 | round | white |
| *Gentiana acaulis* | stemless gentian | *Gentianaceae* | 12 | radial | 4.88 | 4.21 | 3.15 | 4.15 | sharp | blue |
| *Althaea officinalis* | marsh-mallow | *Malvaceae* | 13 | radial | 4.42 | 4.13 | 2.29 | 4.85 | round | white |
| *Geranium palustre* | marsh cranes-bill | *Geraniaceae* | 13 | radial | 4.65 | 4.37 | 1.79 | 5.32 | round | purple |
| *Geum urbanum* | wood avens | *Rosaceae* | 14 | radial | 4.36 | 4.83 | 3.54 | 4.32 | mixed | yellow |
| *Potentilla sterilis* | barren strawberry | *Rosaceae* | 14 | radial | 4.52 | 4.63 | 3.53 | 3.82 | mixed | white |
| *Crepis biennis* | rough hawks-beard | *Asteraceae* | 15 | radial | 4.4 | 4.37 | 2.68 | 5.06 | sharp | yellow |
| *Hieracium aurantiacum* | orange hawk-weed | *Asteraceae* | 15 | radial | 4.59 | 4.15 | 3.55 | 4.38 | sharp | NA |
| *Hypericum perforatum* | St John's wort | *Hypericaceae* | 16 | radial | 4.5 | 4.84 | 2.7 | 4.79 | mixed | yellow |
| *Rubus fruticosus agg.* | blackberry | *Rosaceae* | 16 | radial | 3.63 | 3.7 | 2.56 | 4.72 | mixed | white |
| *Atropa bella-donna* | deadly nightshade | *Solanaceae* | 17 | radial | 3.44 | 3.59 | 3.68 | 2.86 | mixed | NA |
| *Campanula rotundifolia* | harebell | *Campanulaceae* | 17 | radial | 5.05 | 4.87 | 2.6 | 4.5 | sharp | blue |
| *Lathyrus tuberosus* | tuberous pea | *Fabaceae* | 18 | bilateral | 3.66 | 3.14 | 4.34 | 2.31 | round | pink |
| *Pisum sativum* | garden pea | *Fabaceae* | 18 | bilateral | 3.64 | 3.66 | 4.59 | 2.51 | mixed | white |
| *Gentiana verna* | spring gentian | *Gentianaceae* | 19 | radial | 4.82 | 4.12 | 3.15 | 4.02 | round | blue |
| *Silene dioica* | red campion | *Caryophyllaceae* | 19 | radial | 4.27 | 4.12 | 3.57 | 3.72 | round | pink |
| *Viola biflora* | alpine yellow-violet | *Violaceae* | 20 | bilateral | 3.93 | 3.85 | 3.68 | 2.95 | mixed | yellow |
| *Viola reichenbachiana* | early dog-violet | *Violaceae* | 20 | bilateral | 4.09 | 3.57 | 2.81 | 4.14 | round | blue |
| *Borago officinalis* | borage | *Boraginaceae* | 21 | radial | 4.78 | 4.31 | 3.81 | 3.6 | sharp | blue |
| *Swertia perennis* | felwort | *Gentianaceae* | 21 | radial | 4.34 | 4.27 | 3.92 | 3.19 | sharp | blue |
| *Ficaria verna* | lesser celandine | *Ranunculaceae* | 22 | radial | 4.43 | 4.63 | 2.12 | 5.09 | mixed | yellow |
| *Xeranthemum annuum* | immortelle | *Asteraceae* | 22 | radial | 4.7 | 4.44 | 2.33 | 5.02 | sharp | purple |

Hůla and Flegr (2016), *PeerJ*, DOI 10.7717/peerj.2106

**Table 2** (*continued*)

| Scientific name | English name | Family | Pair | Symmetry | Beauty-color | Beauty-sepia | Complexity | Prototypicality | Angularity | Color |
|---|---|---|---|---|---|---|---|---|---|---|
| *Cymbalaria muralis* | ivy-leaved toadflox | *Orobanchaceae* | 23 | bilateral | 3.5 | 3.04 | 4.9 | 2.23 | mixed | blue |
| *Kickxia elatine* | cancerwort | *Orobanchaceae* | 23 | bilateral | 3.21 | 3.04 | 5.47 | 1.64 | mixed | NA |
| *Epipactis palustris* | marsh helle-borine | *Orchidaceae* | 24 | bilateral | 3.86 | 3.74 | 5.18 | 2.15 | mixed | NA |
| *Ophrys apifera* | bee orchid | *Orchidaceae* | 24 | bilateral | 3.7 | 3.5 | 5.28 | 2 | round | pink |
| *Geranium pyrenaicum* | hedgerow geranium | *Geraniaceae* | 25 | radial | 4.72 | 4.64 | 2.39 | 4.99 | round | purple |
| *Stellaria holostea* | greater stitch-wort | *Caryophyllaceae* | 25 | radial | 4.56 | 4.51 | 2.35 | 4.78 | round | white |
| *Arctium tomentosum* | downy bur-dock | *Asteraceae* | 26 | radial | 3.6 | 3.12 | 4.48 | 2.43 | sharp | NA |
| *Cirsium arvense* | creeping thistle | *Asteraceae* | 26 | radial | 3.92 | 3.67 | 4.13 | 2.95 | mixed | purple |

**Notes.**

1, least beautiful/complex/prototypical; 6, most beautiful/complex/prototypical.

# APPENDIX 2. COLOR ANALYSIS—ANOVA TABLES OF THE GENERAL LINEAR MODELS

ANOVA tables and coefficient estimates of the final reduced models are shown. The difference between the mean beauty scores of the colored and sepia tone flowers was used as the dependent variable. All effect remained significant after backward sequential correction for multiple tests. See sections 'Determination of flower traits,' 'Survey design' and 'Flower color.' for details of the explanatory variables.

**Table 3** Color analysis—ANOVA table of the general linear model (all respondents).

|  | df | Sum of squares | F | p-value |
|---|---|---|---|---|
| Hue | 4 | 1.72 | 11.91 | <0.001 |
| Residuals | 41 | 1.48 |  |  |

**Table 4** Color analysis—coefficient estimates of the general linear model (all respondents).

|  | Coefficients estimate | 95% CI | t-value | p-value |
|---|---|---|---|---|
| Intercept (hue = white) | 0.025 | [−0.077–0.13] | 0.49 | 0.62 |
| Hue = yellow | −0.20 | [−0.37–0.02] | −2.35 | 0.024 |
| Hue = pink | 0.20 | [0.026–0.38] | 2.32 | 0.026 |
| Hue = purple | 0.22 | [0.054–0.39] | 2.66 | 0.011 |
| Hue = blue | 0.37 | [0.21–0.54] | 4.61 | <0.001 |

Notes.
Residual standard error, 0.19; df, 41; adjusted $R^2$, 0.49; p-value, 1.64e−06.

**Table 5** Color analysis—ANOVA table of the general linear model (respondents who first rated the sepia tone flowers).

|  | df | Sum of squares | F | p-value |
|---|---|---|---|---|
| Hue | 4 | 1.43 | 10.48 | <0.001 |
| Residuals | 41 | 1.40 |  |  |

**Table 6** Color analysis—coefficient estimates of the general linear model (respondents who first rated the sepia tone flowers).

|  | Coefficients estimate | 95% CI | t-value | p-value |
|---|---|---|---|---|
| Intercept (hue = white) | 0.046 | [−0.05–0.15] | 0.92 | 0.36 |
| Hue = yellow | −0.19 | [−0.35–0.022] | −2.29 | 0.028 |
| Hue = pink | 0.15 | [−0.021–0.32] | 1.77 | 0.084 |
| Hue = purple | 0.21 | [0.047–0.38] | 2.59 | 0.013 |
| Hue = blue | 0.34 | [0.18–0.50] | 4.30 | <0.001 |

Notes.
Residual standard error, 0.18; df, 41; adjusted $R^2$, 0.46; p-value, 6.086e−06.

**Table 7 Color analysis—ANOVA table of the general linear model (respondents who first rated the colored flowers).**

| | df | Sum of squares | F | p-value |
|---|---|---|---|---|
| Hue | 4 | 2.83 | 17.33 | <0.001 |
| Symmetry | 1 | 0.86 | 21.14 | <0.001 |
| Residuals | 40 | 1.63 | | |

**Table 8 Color analysis—coefficient estimates of the general linear model (respondents who first rated the colored flowers).**

| | Coefficients estimate | 95% CI | t-value | p-value |
|---|---|---|---|---|
| Intercept | −0.039 | [−0.15–0.073] | −0.71 | 0.48 |
| Hue = yellow | −0.23 | [−0.41–0.047] | −2.54 | 0.015 |
| Hue = pink | 0.17 | [−0.034–0.37] | 1.68 | 0.10 |
| Hue = purple | 0.29 | [0.11, 0.47] | 3.19 | 0.0028 |
| Hue = blue | 0.46 | [0.29–0.64] | 5.32 | <0.001 |
| Symmetry = bilateral | 0.38 | [0.22–0.55] | 4.60 | <0.001 |

Notes.
Residual standard error, 0.20; $df$, 40; adjusted $R^2$, 0.66; p-value, 2.37e-09.

# APPENDIX 3. SHAPE AND COLOR ANALYSIS—ANOVA TABLES OF THE GENERAL LINEAR MODELS

ANOVA tables and coefficient estimates of the final reduced models are shown. The mean beauty score of the colored flowers was used as the dependent variable. All effect remained significant after backward sequential correction for multiple tests. See 'Determination of flower traits,' 'Survey design' and 'Beauty scores and flower traits. for details of the explanatory variables.

**Table 9 Shape and color analysis—ANOVA table of the general linear model (all respondents).**

| | df | Sum of squares | F | p-value |
|---|---|---|---|---|
| Prototypicality | 1 | 7.48 | 96.37 | <0.001 |
| Hue = blue | 1 | 1.18 | 15.20 | 0.00030 |
| Angularity = sharp | 1 | 0.61 | 7.88 | 0.0072 |
| Residuals | 48 | 3.72 | | |

**Table 10 Shape and color analysis—coefficient estimates of the general linear model (all respondents).**

| | Coefficients estimate | 95% CI | t-value | p-value |
|---|---|---|---|---|
| Intercept | 2.84 | [2.58, 3.11] | 21.74 | <0.001 |
| Prototypicality | 0.31 | [0.24, 037] | 9.30 | <0.001 |
| Hue = blue | 0.35 | [0.14, 0.56] | 3.33 | 0.0017 |
| Angularity = sharp | 0.25 | [0.07, 0.43] | 2.81 | 0.0072 |

Notes.
Residual standard error, 0.28; $df$, 48; adjusted $R^2$, 0.70; p-value, 4.53e-13.

**Table 11  Shape and color analysis—ANOVA table of the general linear model (respondents who first rated the sepia tone flowers).**

|  | df | Sum of squares | F | p-value |
|---|---|---|---|---|
| Prototypicality | 1 | 8.15 | 100.96 | <0.001 |
| Hue = blue | 1 | 1.085 | 13.44 | <0.001 |
| Angularity = sharp | 1 | 0.62 | 7.68 | 0.0079 |
| Residuals | 48 | 3.87 |  |  |

**Table 12  Shape and color analysis—coefficient estimates of the general linear model (respondents who first rated the sepia tone flowers).**

|  | Coefficients estimate | 95% CI | t-value | p-value |
|---|---|---|---|---|
| Intercept | 2.80 | [2.53–.069] | 20.99 | <0.001 |
| Prototypicality | 0.32 | [0.25–0.39] | 9.56 | <0.001 |
| Hue = blue | 0.33 | [0.12–0.55] | 3.11 | 0.0032 |
| Angularity = sharp | 0.25 | [0.069–0.43] | 2.77 | 0.0079 |

**Notes.**
Residual standard error, 0.28; df, 48; adjusted $R^2$, 0.70; p-value, 3.13e-13.

**Table 13  Shape and color analysis—ANOVA table of the general linear model (respondents who first rated the colored flowers).**

|  | df | Sum of squares | F | p-value |
|---|---|---|---|---|
| Prototypicality | 1 | 5.66 | 78.85 | <0.001 |
| Hue = blue | 1 | 1.54 | 21.52 | <0.001 |
| Angularity = sharp | 1 | 0.47 | 6.60 | 0.013 |
| Residuals | 48 | 3.44 |  |  |

**Table 14  Shape and color analysis—coefficient estimates of the general linear model (respondents who first rated the colored flowers).**

|  | Coefficients estimate | 95% CI | t-value | p-value |
|---|---|---|---|---|
| Intercept | 2.96 | [2.71–3.22] | 23.56 | <0.001 |
| Prototypicality | 0.26 | [0.20–0.33] | 8.29 | <0.001 |
| Hue = blue | 0.41 | [0.21–0.61] | 4.10 | <0.001 |
| Angularity = sharp | 0.22 | [0.047–0.39] | 2.57 | 0.013 |

**Notes.**
Residual standard error, 0.27; df, 48; adjusted $R^2$, 0.67; p-value, 2.86e-12.

## ACKNOWLEDGEMENTS

We would like to thank Anna M. Borghi, Dermot Lynott and an anonymous reviewer for their invaluable comments and suggestions which helped us to increase the quality of the present paper. We also thank Jim Dutt and Raymon Gongora for language help.

### Funding

The research was supported by the Grant Agency of the Charles University in Prague (project no: 388315). The funders had no role in study design, data collection and analysis, decision to publish, or preparation of the manuscript.

### Grant Disclosures

The following grant information was disclosed by the authors:
Grant Agency of the Charles University in Prague: 388315.

### Competing Interests

The authors declare there are no competing interests.

### Author Contributions

- Martin Hůla conceived and designed the experiments, performed the experiments, analyzed the data, contributed reagents/materials/analysis tools, wrote the paper, prepared figures and/or tables, reviewed drafts of the paper.
- Jaroslav Flegr conceived and designed the experiments, analyzed the data, contributed reagents/materials/analysis tools, reviewed drafts of the paper.

### Human Ethics

The following information was supplied relating to ethical approvals (i.e., approving body and any reference numbers):
Charles University, Faculty of Science IRB approval number: 2015/31.

### Data Availability

Figshare: 10.6084/m9.figshare.2082529; https://figshare.com/s/7306f12659f68f7f3d9d.

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
