# Peer review of "What flowers do we like? The influence of shape and color on the rating of flower beauty"

_PeerJ, doi:10.7717/peerj.2106_

## Round 0.1 · original submission · Major Revisions

Dear Dr. Hula,

I write regarding your manuscript, submitted to PeerJ. Attached below you will find the comments from two reviewers I asked to evaluate your work. I thank the reviewers for their work.

As you will see, the reviewers profoundly diverge in their overall assessment of the paper: while reviewer 2 (Dermot Lynott) is optimistic about the eventual publishability of the work (minor revision), reviewer 1 proposes to reject the paper. On the basis of my own independent reading of your paper and of the reviewers reports, I have decided to give you a chance to review this manuscript for further consideration on this journal. I see some categories of issues that I consider crucial for a successful revision:

A. Theoretical framing. The major limitation of the work is that its theoretical underpinnings should be better fleshed out (see comments of both reviewers). The authors refer in the discussion to theories of habitat selection; they might want to start discussing this theories ealier in the introduction, to motivate their study. The psychological literature on shape, color and on the relationship between shape and color during objects and entities recognition and representation can also be useful to strengthen the theoretical part of the paper (e.g. Bramao et al., 2011; Connel & Lynott, 2009; Morlino et al., 2015; Scorolli & Borghi, 2015; Therriault et al., 2009; many studies by Linda Smith, e.g. Smith 2005).

B. Aim of the study. The authors should better clarify whether they intend to investigate mechanisms underlying beauty perception in general or beauty of flowers (see comments of reviewer 1).

C. Design. I agree with reviewer 2 that sepia judgements could be lowered because of participant’s judgement of colored images. The authors should address this issue in their revision.

I invite you to carefully address all the reviewers' comments. If you decide against implementing some suggestions, please explain the rationale in your rebuttal letter. Thank you for submitting your interesting work to PeerJ.

Sincerely
Anna Borghi

Reviewer 1 ·

Basic reporting

In this study the authors investigate what flowers do humans like. Participants completed two online questioners where they rated how beautiful they consider each flower and the prototypicality and complexity of each item. The authors investigated which properties contributed for the overall assessment of beauty and found that participants preferred blue flowers, with symmetrical and simple shapes.

Experimental design

The theoretical motivation for conducting the study is lacking. This is an exploratory study, where the research question is not supported by a clear and relevant theoretical background.

Validity of the findings

If the question is to answer which properties contributes to for the beautiful perception of flowers, a control condition should have been included, in order to draw specific conclusions about the flowers. Such a control condition is absent. The way it is presented now, the study does not allow to draw specific conclusions about the beauty perception of flowers. The results can just be a reflection of the properties that are important for the beauty perception in general. Also, the previous knowledge about the items will determine how beautiful an item will be perceived. Controlling for the previous knowledge with the items is also important. This can be done by having participants rating non-objects shapes, where no previous knowledge about the items could contribute for the beauty perception.

·

Basic reporting

Basic reporting is clear and detailed, and the submission is self-contained. There is possibly no need for ANOVA tables in the main text.

Experimental design

The aim of the research is clear, and the authors highlight gaps in research on aesthetic judgements which motivates the current work. As this this work is outside my area, I can only accept this at face value. The methods are clearly described, along with selection criteria, procedure and analytical choices. There are several hypotheses, clearly made. Where sometimes it seemed that the background literature could support or contradict some of these predictions, the authors are clear on the lack of a directional prediction (e.g., bilateral vs radial symmetry).
I have one concern about the design that may be worth considering. Because participants saw both colour and sepia images, it is possible that this contrast lead people to rate the colour flowers more positively overall. In other words, sepia judgements were lowered because of participant’s exposure to and judgement of colour images.
It’s an empirical question, but I suspect if people judged only sepia images you would see an increase in the variance of ratings of the sepia images, relative to the current study.

Validity of the findings

The analysis seems appropriate and thorough. In one or two places in the manuscript I have asked for additional details. The conclusions are appropriate, although I think it would be nice to flesh out in the discussion the specific aspects of the findings that support the referenced theories of habitat selection ecological valence theory. Also, are there any aspects of the findings that count against particular theoretical perspectives?

---

## Round 0.2 · accepted · Accept

I am happy to inform you that your paper has been accepted for publication on PeerJ. I think it represents an interesting contribution to the studies aimed at understanding the basis of beauty perception.
Congratulations, and thank you for your submission.
Anna

·

Basic reporting

n/a

Experimental design

n/a

Validity of the findings

n/a

Additional comments

The authors have addressed my previous comments. I think there probably is scope for deeper theoretical analysis of the various evolutionary accounts for the observed preferences, but equally there is probably sufficient discussion for this empirical paper.